# Microwave Receiving System Based on Cryogenic Sensors for the Optical Big Telescope Alt-Azimuth

**DOI:** 10.3390/s24020359

**Published:** 2024-01-07

**Authors:** Yurii Balega, Grigory Bubnov, Artem Chekushkin, Victor Dubrovich, Valerian Edelman, Aleksandra Gunbina, Sergey Kapustin, Tatyana Khabarova, Dmitrii Kukushkin, Igor Lapkin, Maria Mansfeld, Andrei Maruhno, Vladimir Parshin, Aleksey Raevskiy, Vladislav Stolyarov, Mikhail Tarasov, Gennady Valyavin, Vyacheslav Vdovin, Grigory Yakopov, Renat Yusupov, Petr Zemlyanukha, Igor Zinchenko

**Affiliations:** 1Special Astrophysical Observatory RAS, Nizhnii Arkhyz 369167, Russia; balega@sao.ru (Y.B.); dubrovich@sao.ru (V.D.); s.kapustin@ipfran.ru (S.K.); dekukushkin@itmo.ru (D.K.); marym@ipfran.ru (M.M.); mas@sao.ru (A.M.); vlad@sao.ru (V.S.); gvalyavin@sao.ru (G.V.); yakopov@sao.ru (G.Y.); 2A.V. Gaponov-Grekhov Institute of Applied Physics RAS, Nizhniy Novgorod 603950, Russia; payalnik89@gmail.com (G.B.); gunbina@ipfran.ru (A.G.); t.habarova@ipfran.ru (T.K.); lapkin@ipfran.ru (I.L.); parsh@ipfran.ru (V.P.); petez@ipfran.ru (P.Z.); zin@ipfran.ru (I.Z.); 3The Kotel’nikov Institute of Radio Engineering and Electronics of RAS, Moscow 125009, Russia; chekushkin@hitech.cplire.ru (A.C.); tarasov@hitech.cplire.ru (M.T.); yusupovrenat@hitech.cplire.ru (R.Y.); 4P.L. Kapitza Institute for Physical Problems RAS, Moscow 119334, Russia; edelman@kapitza.ras.ru; 5Department of Physics and Technology of Optical Communication, Nizhny Novgorod State Technical University n.a. R.E. Alekseev, Nizhniy Novgorod 603155, Russia; raevsky@nntu.ru; 6Applied Optics Center, ITMO University, Saint Petersburg 197101, Russia

**Keywords:** radioastronomy, superconducting sensors, SINIS detectors, optical telescope, BTA SAO RAS

## Abstract

This article presents the results of evaluating the possibility of conducting radio astronomy studies in the windows of atmospheric transparency ~100, ~230, and ~350 GHz using the optical Big Telescope Alt-Azimuthal (BTA) of the Special Astrophysical Observatory of the Russian Academy of Sciences (SAO RAS). A list of some promising astronomical tasks is proposed. The astroclimat conditions at the BTA site and possible optical, cryogenic, and mechanical interfaces for mounting a superconducting radio receiver at the focus of the optical telescope are considered. As a receiving system, arrays of detectors cooled to ~0.3 K based on the superconductor–insulator–normal metal–insulator–superconductor (SINIS) structure are proposed. The implementation of the project will make it possible to use the BTA site of the SAO RAS not only to solve some astronomical problems (it is possible to consider the implementation of a single observatory, the VLBI (very-long-baseline interferometry) mode in the Suffa, EHT (Event Horizon Telescope), and Millimetron projects), but it will also be used to test various cryogenic detectors in a real observatory.

## 1. Introduction

Microwave radio astronomy has been actively developing since the 1960s (an overview of this stage of development is presented, for example, in [1]). The first large antennas of this range were built, for example, the 11 m radio telescope of the NRAO USA, RT-22 LPI (P.N. Lebedev Physical Institute of the Russian Academy of Sciences), RT-22 KrAO (Crimean Astrophysical Observatory), etc. Quite sensitive uncooled detectors have been created, followed by cryogenic detectors (as an example, let us cite such leading concepts as a Transition Edge Sensor (TES) [2,3,4,5,6] and Kinetic Inductance Detectors (KIDs) [7,8,9,10]. In subsequent years, impressive progress has been observed in improving the reception technology of millimeter waves, stimulated to a large extent precisely by the needs of radio astronomy. The noise temperature of receiving systems—the main parameter that characterizes their sensitivity—has decreased by about two orders of magnitude since the late 1960s and has come close to the so-called quantum limit. Many new ground-based [11,12,13,14] balloon [15,16,17] and space [18,19] radio telescopes of this range have been built, including several antenna arrays. 

Currently, there are many important astrophysical problems that can be solved by millimeter (MM) and sub-MM astronomy [20,21,22]. Many of them are related to the study of the “cold” Universe. These are mainly dense interstellar gas–dust clouds, both in our galaxy and in distant objects. They are interesting, first of all, because the process of star formation takes place in them, many aspects of which are still not fully understood. The temperature of these clouds is from several to tens of Kelvins. The peak of their emission at this temperature is in the MM-sub-MM range. This range is also extremely rich in spectral lines, mainly corresponding to transition between rotational levels of molecules with relatively low excitation energy. A lot of various molecules have been found in interstellar clouds. In addition, lines are observed that arise during transitions between levels of the fine structure of some atoms and ions. Observations of spectral lines make it possible to study the physical conditions and processes in interstellar clouds, as well as their chemical composition. In addition, the heating of dust and gas is carried out by ultraviolet radiation from stars and other factors related ultimately to the processes of energy release in protostars and young stars. It is important that interstellar clouds are practically transparent for MM and sub-MM waves, in contrast to the optical range, where absorption can reach 100^m^ or more. Thus, observations in this range represent a unique tool for studying the inner regions of dense interstellar clouds, which are the “cradles” of new stars and are inaccessible to other methods of astronomical research. Studies of the formation of stars and galaxies are one of the main incentives for the development of millimeter and submillimeter astronomy. The “cold” objects of study also include the microwave “relict” background, the peak intensity of which falls in the millimeter range.

Along with the study of the cold universe, MM and sub-MM (or sub-THz) astronomy makes an invaluable contribution to the study of objects characterized by extremely high energy release, such as active galactic nuclei, which, apparently, are accreting supermassive black holes. It is in this range that it is possible to realize the required ultra-high angular resolution, as well as reduce the influence of scattering effects along the line of sight.

The sensors and receivers of sub-THz waves presented in this work are designed to solve the problems of sub-THz astronomy listed above. However, they require radio telescopes of the appropriate range and good conditions for the propagation of these waves in the atmosphere. There is very large absorption due to interaction with molecules of air, oxygen, and water vapor in the first place. The combination of atmospheric factors is called microwave astroclimate below. And the location of telescopes is usually characterized by high altitudes above sea level, where there is little oxygen, and the water is frozen due to low temperatures. For sub-THz astronomy tasks, the densest coverage of the Earth with telescopes is desirable. However, it so happens today that there are practically no such instruments in northeastern Eurasia. And even if there are surfaces of suitable quality (RMS no worse than tens of microns), they do not have a good astroclimate. Only RT-13 IAA RAS (Institute of Applied Astronomy Russian Academy of Science) [23] has a surface accuracy sufficient for effective operation at wavelengths of ~1 mm. The quality of the 7.5 m antennas of the Bauman Moscow State Technical University in Dmitrov (Moscow region) is positioned slightly worse [24]. However, the astroclimate in their locations will most likely not allow them to be used effectively at wavelengths shorter than ~2 mm. Russia’s entry into the European Southern Observatory could partially correct this situation; however, despite many years of efforts being made in this direction, the prospects for this remain very vague, and over the past year, they seem to have been completely lost for a long time. Two instruments here are under construction now; the central place is occupied by the International Russian–Uzbek megaproject “Suffa” for the construction of an observatory [25] on the Suffa plateau in Uzbekistan and space observatory Millimetron both designed by Astrospace center P.N. Lebedev PI RAS. But both of them are unlikely to be built this decade. There are some already built telescopes in Korea, but there is the problem of astroclimat here. In China, there are lot of excellent locations and big projects. But the QTT project, which is closest to our goals, has been frozen since 2018. The BTA optical telescope (Big Telescope Alt-Azimuth, (see Figure 1) in the SAO RAS [26] under the circumstances may well be used now to discover techniques and receive devices of this range (as well as a number of urgent tasks of modern MM and sub-MM astronomy). The surface quality of the optical telescope mirror will certainly (by two orders of magnitude) satisfy the requirements for research in the sub-terahertz range. Observations in MM and sub-MM ranges can be carried out in the early morning or late evening, in contrast to observations in the optical range, which are carried out only at night in the absence of background illumination of the sky.

Potentially, daylight time, which is not used for optical observations, is also quite suitable for MM observations, but opening the visor of the telescope dome in the daylight time, as a rule, is not allowed from the point of view of the thermalization of the mirror and the dome space, which is actively stabilized for night mode. In the afternoon, after sunrise, dramatic thermal processes are possible, calling into question the success of thermalization in the evening, which will negatively affect the quality of subsequent night optical observations. Thus, expanding the functionality of the telescope into the MM region is some addition to the functionality and the main observation program and does not interfere with it, unless the dome is opened significantly beyond the twilight period. This is an absolute advantage of the proposed scheme for supplementing the BTA with an MM receiver.

## 2. Astroclimate Conditions at the BTA Site of the SAO RAS

To select the frequency range of the receiving system, it is necessary to assess the astroclimate conditions in the windows of atmospheric transparency. An active study of the astroclimate in the vicinity of the BTA was carried out with the participation of the authors of this article; a review of the results of the astroclimate measurements taken at various sites is given in [27] and a series of papers cited there.

To monitor the transparency of the atmosphere, the MIAP-2 device (a two-channel microwave atmospheric absorption meter) was used, described in some detail in [28], as well as a lot of connected publications. Its block diagram and photo can be seen in Figure 2. The methodology for processing the measured data is given in [27] and some other publications cited there.

Some results of the investigation of the propagation of sub-THz waves in atmosphere for transparency windows of 3, 2, 1.3, and 0.8 mm are presented in [28] and some following papers. The results are shown in Figure 3, and histograms by month and season are outlined in Figure 4.

The key conclusion from long-term cycles of observations of the microwave astroclimate directly from the BTA telescope (photo of the device on the window, Figure 2b) is as follows: observations in the (~100 and ~150 GHz) transparency windows undoubtedly do not have significant atmospheric restrictions, and observations in the transparency window of ~230 GHz require the selection of certain periods of time; clear, frosty winter days are quite suitable. The use of the BTA in subsequent (350 GHz and above) windows of atmospheric transparency is practically unpromising due to the high humidity of the atmosphere and the insufficiently high altitude of the telescope above sea level.

The conclusion presented here, obtained on the basis of our direct measurements of the absorption of sub-THz waves, agrees well and is complemented by estimates made by indirect methods [29,30,31] measured near the site of the BTA telescope.

## 3. Receiving System for the BTA

Despite the extensive list of current astronomical tasks in the MM frequency range, there are various limitations of a ready-made instrument (especially for the BTA designed for the optical range). Therefore, in this section, a preliminary assessment will be made of the possibilities of creating and mounting an MM receiver on the site of the BTA telescope, and then, based on the results obtained, a set of relevant astronomical problems proposed for research based on the optical telescope will be considered.

Initially, we propose to study the surface of the telescope mirror (see Section 3.1). If the surface quality (roughness value) of the optical mirror certainly meets the requirements of the MM range, then the aluminum coating (its thickness is less than the skin layer in this wavelength) requires additional research.

Another issue is the proposed location of the receiving system and the estimation of the width of the Gaussian beam (detailed in Section 3.2). As can be seen from Figure 1, there are three possible locations for the detection module: primary focus and I or II Nasmith focus. Setting it to primary focus is impossible because this will interfere with the telescope’s main program. However, installing a cooled receiver in the Nasmith II or I circuit is a real possibility.

From 23 October to 29 October 2019, an expedition was conducted to the Special Astrophysical Observatory of the Russian Academy of Sciences to discuss collaboration on the installation of an MM receiving system. Some photographs are shown in Figure 5.

### 3.1. Surface of the BTA’s Mirror

The mirror of the BTA telescope is a monolithic glass with a diameter of 605 cm with an aluminum coating of ~100 nm and a weight of about 42 tons. In 2018, the surface of the telescope’s main mirror was replaced (redeposition) [32]. Special vacuum equipment located inside the observatory was used to cover the surface of the mirror with aluminum. In the process of the deposition of the main mirror, special “witness” samples were made, namely test samples that were deposited in the same vacuum cycle with the main mirror. They are necessary for studying the characteristics of the mirror surface. Experimental studies of the reflection losses of two samples of “witnesses” were conducted on the basis of the IAP RAS (Figure 6). The sample (Figure 6a) is a polished glass disk with a diameter of 30 mm with a 100 nm thick aluminum film deposited on it. Reflectivity measurements were carried out in a cryovacuum resonator complex [33]. The experimental stand from [33] is based on a high-quality (Q~10^6^) Fabry–Perot resonator, which is located in a vacuum chamber. A symmetrical resonator was used as a reference, and a test sample of the reflector was installed on a flat copper mirror of an asymmetric resonator. Reflection losses from samples at normal radiation incidence were determined based on the results of the measurements of the resonator response width.

According to the measurement results (Figure 6b), it can be seen that the measured reflection loss is 5–6% in the frequency range 120–200 GHz at room temperature. The efficiency of the mirror as an MM and sub-MM wave reflector (better than 94%) is no worse than the measured reflection coefficient in the “native” optical range (400–1200 nm [Figure 2 and Figure 3 in [32]]). 

### 3.2. The Nasmith System of the BTA Telescope

A schematic representation of the Nasmith I system of the BTA telescope is shown in Figure 7. The signal from the main mirror reaches the secondary mirror, and then the focusing, deflecting the beam mirror to the first or second Nasmith focus of the telescope. Two methods for calculating the width of the Gaussian beam will be considered: quasi-optical and using standard software for optical calculations, Zemax OpticStudio.
**Quasi-optical calculation method**

The calculation method is given in detail in [34].
*Calculation of the Gaussian beam radius in the Nasmith 1 scheme of the BTA telescope*

The matrix of the propagation of a Gaussian beam through two free spaces and a mirror:(1)$$M=\left(\begin{array}{cc} 1 & L_{2}\\ 0 & 1 \end{array}\right)\cdot \left(\begin{array}{cc} 1 & 0\\ {-F}_{mirr}^{-1} & 1 \end{array}\right)\cdot \left(\begin{array}{cc} 1 & L_{1}\\ 0 & 1 \end{array}\right)=\left[\begin{array}{cc} 1-\frac{L_{2}}{F_{mirr}} & \left(1-\frac{L_{2}}{F_{mirr}}\right)\cdot L_{1}+L_{2}\\ -\frac{1}{F_{mirr}} & -\frac{1}{F_{mirr}}\cdot L_{1}+1 \end{array}\right]$$
*L*_1_ is the distance from the aperture of the horn to the refocusing mirror;*F_mirr_* is the focal length of the focusing mirror;*L*_2_ is the effective focal length of the main mirror.

(2)$$F_{mirr}=h_{f}\cdot R_{s}\cdot \frac{1-K}{\left(-h_{f}\right)\cdot K^{2}+2\cdot h_{f}\cdot K-h_{f}+R_{s}}$$
where $K=1+\frac{w_{s}}{h_{a}\cdot k}$ is an auxiliary coefficient (*w_s_* is the radius of the Gaussian beam on the secondary mirror, *h_a_* is the size of the aperture of the horn, and *k* is the Gaussian coefficient).



(3)
$$L_{1}=F_{mirr}\cdot \left(\frac{h_{a}\cdot k}{w_{s}}+1\right)$$


(4)
$$L_{2}=F_{mirr}\cdot K$$



Beam parameters:(5)$$q_{in}={\left[\frac{1}{h_{f}}-i\frac{\lambda }{\pi \cdot {\left(a_{h}\cdot k\right)}^{2}}\right]}^{-1}$$
(6)$$q_{out}=\frac{\left(M_{0,0}\cdot q_{in}+M_{0,1}\right)}{\left(M_{1,0}\cdot q_{in}+M_{1,1}\right)}$$*h_f_* is the magnitude of the curvature of the phase front after the horn (∞).

The radius of the Gaussian beam during propagation in the Nasmith I system of the BTA telescope:(7)$$w\left(z\right)=\sqrt{\lambda \cdot {\left[\pi \cdot Im({-\left(q_{out}+z\right)}^{-1})\right]}^{-1}}$$*z* is the Gaussian beam propagation distance (from 0 to 22,014 mm).

The calculated radius of a Gaussian beam propagation in the Nasmith I scheme of the BTA telescope of the SAO RAS is presented in Figure 8.
*Losses during propagation of a Gaussian beam through the “tube”*

When propagating a Gaussian beam in the Nasmith I scheme, to determine the losses when passing through the “Nasmith tube”, it is necessary to consider the truncation of the beam, and the following formula can be used [34]:(8)$$f_{out}=exp\left[-2{\left(\frac{r_{t}}{w^{2}}\right)}^{2}\right]$$
where *r_t_* is the radius of the beam truncation (in our case, it is equal to the radius of the pipe—110 mm). Accordingly, the share of passing power in % is (1 − *f_out_*) × 100.
**Calculation in Zemax OpticStudio**

Standard software for optical calculations, Zemax OpticStudio, was also used to estimate the beam size in the Nasmith focus of the BTA telescope. Despite the fact that the operating spectral range of the detector goes beyond the limits of the optical radiation range, the analysis of the diffraction image diameter can be performed using the wave effect analysis tools in Zemax OpticStudio (PSF, MTF, etc.) [35,36]. Figure 9 shows graphs of Huygens PSF scattering spots in the XY plane at the Nasmith focus for the wavelengths λ = 1303 μm (230 GHz) and λ = 2998 μm (100 GHz). The Huygens PSF method was used because it is more accurate and takes into account the discrepancy between the position of the planes of the geometric focus and the image of the optical system. The diameter of the first diffraction maximum at λ = 1303 μm (230 GHz) made up D(λ = 1303 μm)1-st dif.max. ≈ 88 mm, at λ = 2998 μm (230 GHz) D(λ = 2998 μm)1-st dif.max. ≈ 200 mm. The results of estimating the Gaussian beam radius in Zemax OpticStudio using the “Gaussian paraxial beam” with a Gaussian beam apodization coefficient k = 0.644 are presented in Table 1.

This section provides information only to determine the possibility of a Gaussian beam passing into the Nasmith I focus of the BTA telescope and the need to create corrective elements before arriving at the Nasmith tube at frequencies of 90 and 230 GHz (as shown in the calculations above, it is not required, which greatly simplifies the problem since corrective elements will only be required after passing through the Nasmith tube). For more detailed calculations and calculations of additional elements for supplying a signal to the detecting device, it is necessary to clarify the following points: Selection of an astronomical task (depending on this, the central frequency of the signal and the reception bandwidth will be determined, as well as the choice of matching horn, lens, mirror, or horn–lens (mirror) structures);Selection of a cryogenic system and determination of its exact overall dimensions (depending on the size and location of the cryostat system, additional mirrors for “transferring” the signal to the cryostat window will be calculated);Determination of the maximum permissible diameter of the window in the cryostat through which the signal will be supplied to the detecting device (depending on this size, corrective elements for beam narrowing will be modeled).

Since the maximum diameter of the cryostat window is significantly smaller than the beam diameter in the Nasmith focus, it was planned to use a matching optical system of off-axis parabolic mirrors for the scaling of the output numerical aperture of the telescope. The relative aperture of the telescope in the Nasmith focus is 1/30 (F = 30), which corresponds to the diameter of the first diffraction maximum for λ = 2998 μm D1-st dif.max. ≈ 200 mm. Preliminary calculations of the matching optics showed that the optimal diameter of the horn is 20 mm. Hence, the linear magnification of the optical system of two paraboloids is V = $f_{2}^{’}/{f_{1}^{’}=0.1}^{\times }$. The main limitation of the maximum achievable increase in matching optics is the dimensions of the Nasmith platform and the design features of the receiver. The image size depends on the ratio of the focal lengths of the paraboloids. An increase in the focal length of the first paraboloid will lead to an increase in the diameter of the collimated beam and the length of the system; a decrease in the focal length of the second paraboloid will lead to a decrease in the rear segment of the system, which will not allow the cryostat horn to be placed in the correct position.

### 3.3. Cryogenic System

The sensors presented in this article have ultra-high sensitivity, and a prerequisite for achieving it is deep cryogenic cooling of the detectors to sub-Kelvin (subK) temperatures. Such systems are two orders of magnitude lower in temperature than widespread commercial cryogenic systems of hydrogen temperature levels [37] and two orders of magnitude more complex and expensive. Helium temperature level systems [38] provide an order of magnitude higher temperature level and are used in cryosystems used for SINIS sensors for precooling up to 4 K.

Initially, two options were considered as a cryostat system, as shown in Figure 10, namely a system based on pulse tubes and a filler cryostat insert, on which experimental studies of the developed samples were carried out (these cryosystem is used to conduct experimental samples of detecting arrays based on SINIS detectors in the laboratory).

1. Cryostat with pumping out liquid helium vapor 3He Heliox AC-V (Figure 10a). The achievable minimum operating temperature is 273 mK. The advantages of such a cryogenic system include the following: the presence of a closed-cycle cryocooler on pulse tubes, i.e., there is no need to pour liquid helium (but this is also the disadvantage of such a system—the presence of “parasitic” noise from the pulsation system), a large volume for mounting samples (the pad diameter is 147 mm, the height is 100 mm), and high operating temperature stability (±3 mK at temperatures below 2 K).

2. Filler cryostat insert (Figure 10b) that was developed at the P.L. Kapitsa Institute of Physical Problems of the Russian Academy of Sciences [39]. The minimum achievable operating temperature is 80 mK. In addition to achieving an ultra-low temperature, such a cryostat does not require external pumps, which significantly reduces vibration noise and interference (which increases the sensitivity of the entire receiving system); however, this requires pouring liquid helium, and the cycle of continuous operation at a temperature of 0.1 K is quite short.

Both options can be used depending on the chosen task for research at the BTA telescope. The main difference between two fundamentally different approaches when choosing a pre-cooling system for a sub-Kelvin refrigerator to 4 K is that refrigerators are much more convenient without requiring the filling of a cryoagent. And the LHe system is distinguished by the complete absence of vibrations, which creates serious problems when using refrigerators [38] with an essential level of vibrations. Refrigerated or liquid pre-cooling systems provide a preliminary level of 4 K, on the basis of which sub-Kelvin cooling systems are built based on dissolution cryostats or sorption cycles [39,40]. Currently, various options for creating various cryosystems are being considered. The details of these systems are given in [40]. See Section 3.4.

### 3.4. Detecting Device

As a highly sensitive receiving device, the implementation of a receiving cell based on detectors of the superconductor–insulator–normal metal–insulator–superconductor structure is proposed (a schematic image is shown in Figure 11). The sensitive element of such a structure is a submicron absorber made of normal metal (for example, depending on the available technology, aluminum with a sublayer of iron, copper, hafnium, palladium, etc., can be used), which is heated by incoming electromagnetic radiation. Two tunnel junctions of the normal metal–insulator–superconductor (NIS) structure act as thermometers. As it was theoretically shown in [41], SINIS detectors can operate in different modes (depending on the chosen topology), from a photon counter to a bolometric mode with high quantum efficiency. For example, [42] presents the implementation of a SINIS detector with a quantum efficiency of 15 electrons per quant of radiation with a frequency of 350 GHz. The saturation power of a single SINIS detector is of the order of 1 pW. By combining such structures into arrays, it is possible to significantly increase the saturation level of the detecting cell, which is fundamentally important for working in ground-based observatories in conditions of high background load (tens of picowatts). Typical characteristics of the arrays of such detectors are volt–watt sensitivity up to 10^9^ V/W, and noise equivalent power is no worse than 10^−16^ W/$\sqrt{Hz}$ (and can be improved by replacing the warm readout electronics with frequency multiplexing with a cold HEMT amplifier). The frequency response of detecting devices based on SINIS structures is determined by the antenna/array of antennas into which detectors are integrated. In detail, the principle of operation of such detectors is given in [42]. Some typical measured characteristics of SINIS detectors are shown in Figure 12.

For the fabrication of such detection devices, various technologies can be used (depending on the topology and available technological equipment): shadow evaporation technique, bridge-free technology, technology with separate lithography, and magnetron sputtering. In detail, different methods and approaches used to fabricate such structures are given in [43], and photographs of SINIS bolometers fabricated using various technologies are shown in Figure 13.

**Figure 12 sensors-24-00359-f012:**
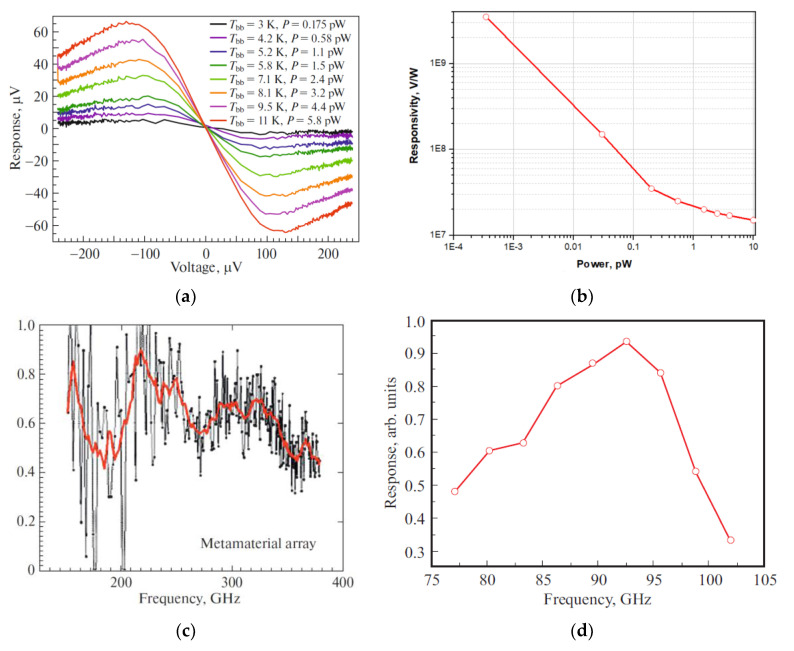
Typical measured characteristics of SINIS detectors: (**a**) voltage responses of a single SINIS detector with a suspended absorber at a temperature of 120 mK to black body radiation of different power [42]. (**b**) Responsivity dependence on power for single bolometer in the series array measured at 100 mK with cryogenic blackbody source (graph from the publication [44]); (**c**) spectral response of an array of electrically small antennas with integrated SINIS detectors of the 350 GHz range [44]; (**d**) spectral response of a single double-slot antenna of the 90 GHz band with an integrated SINIS detector [45].

SINIS detectors themselves are non-selective structures; the frequency range and bandwidth are primarily determined by the planar antennas into which they are integrated, as well as by the nodes of the quasi-optical receiving path, various filtering elements, etc., installed on the source-detecting cell path. Due to the miniature size of the detectors, they can be easily integrated into various planar structures: single antennas, various frequency-selective surfaces, electrically small antennas, etc. Accordingly, there is no problem of switching to a higher frequency range: 1.3 and 0.8 mm (not only for the BTA, but also with the prospect of installation on instruments specifically designed for the sub-THz range).

The assessment of the prospects and possibilities of working in these transparency windows will be carried out after the next cycle of atmospheric absorption research. But if we consider the prospects of using the BTA as part of the EHT, then it should be kept in mind that the SINIS structures are not suitable for an interferometer; the SIS receivers actively developed at the IRE RAS are suitable here (for example, presented in [46,47]).

As noted above, for primary test studies, the 90 GHz range is assumed, which is the most convenient from the point of view of astroclimate conditions (observations can be carried out at any time of the year). Publication [48] presents the results of modeling arrays of receiving annular antennas in the 90 GHz range with irradiation from the substrate side and from the antenna side (Figure 14). The topology of such a structure was developed with two types of connection of array elements (serial and parallel) (Figure 15). Different types of connection of elements provide different resistance: about 36 kOhm and 15 Ohm, respectively. Different resistances of receiving matrices are considered for the possible use of different readout systems: high-resistance based on JFET transistors or low-resistance with SQUID amplifiers.

To match the incoming radiation with the receiving array, it is proposed to use the back-to-back horn. The choice of this design is due to the following: the best way to “cut” the frequency from below (in this case, 75 GHz) is to use a waveguide of a given diameter. The diameter of the waveguide with a critical frequency of 75 GHz is 2.34 mm. The developed receiving array fits into a circle with a diameter of 9.2 mm, and it is impossible to place it directly in the waveguide. In this regard, an additional “transmitting” horn is used, located after the waveguide. A schematic representation of the back-to-back horn with dimensions and its radiation pattern is shown in Figure 16.

### 3.5. Possible Relevant Astronomical Tasks in the sub-THz Range for the BTA Telescope

Possible actual tasks at the BTA site:

The range is about 3 mm (convenient from the point of view of the astroclimate—studies can be carried out all year round). Observations can be carried out for such tasks as:Measurements of the fluxes of bright objects in the continuum by joint programs with the RATAN-600 radio telescope;Studies of active galactic nuclei (AGN).

Millimeter and submillimeter astronomy make an invaluable contribution to the study of objects characterized by extremely high energy release, such as active galactic nuclei, which, apparently, are accreting supermassive black holes.
Studies of lacertids (objects of the BL Lac type);Studies of distant quasars (for example, ULIRG QSO);Conducting observations of extragalactic objects and molecular clouds in our galaxy in the CO J1-0 115.27 GHz line;Estimation of the fluxes of very distant galaxies recently discovered by the JWST Space Telescope (joint optical observation programs with BTA are possible).

## 4. Conclusions

This article proposes a project to use the capabilities of the BTA of the SAO RAS for radio astronomical observations in the MM and sub-MM range, which will potentially expand the capabilities of the BTA optical telescope. The results of the preliminary assessments and studies presented in this article showed that the implementation of such a project is possible, and the team of co-authors of this work demonstrated here their readiness to solve the rather unusual task of the incorporation of a radio wave receiver as the sensor into the BTA optical telescope of the SAO RAS. This site can be used quite effectively both for testing receiving systems under the conditions of a real ground-based instrument and for conducting radioastronomical research on some current astrophysical problems of millimeter-wave astronomy, including to be an element of EHT.
This article is in memory of our departed colleagues who make up the history of this project of mounting sub-THz receivers on the site of the BTA optical telescope of the SAO RAS.

In reality, the history of this project proposed for implementation in cooperation between SAO RAS, IAP RAS, IRE RAS, and IPP RAS has a fairly long duration and has been extremely productive and fruitful. Initially, the idea was formulated, and the first attempt was made to install an uncooled DBS receiver back in the late 1980s–early 1990s. With the help of a joint team, three of whom are the authors, a relatively successful experiment in installing a DBS receiver was carried out. But it was not possible to measure real astronomical objects; it was not clear whether the problem of quasi-optics was successfully solved, whether a glass optical mirror coated with a metal layer obviously thinner than the skin layer at mm waves reflects the mm signal well enough, etc. In the same years, the joint work of IRE and IAP was actively and successfully developed on the basis of the extensive competencies of IRE in superconducting receivers and IAP in cryogenic astronomical mm devices in the creation of superconducting astronomical receivers. As a result, the first domestic receiver was launched on the RT-25*2 telescope in Zimenki near Nizhny Novgorod in 1991 under the leadership of Prof. Albert Kislyakov. But the dramatic turning point of 1991 prevented the continuation of these works due to a lack of funding for them. However, the united team has found new directions and means. The IAP team was invited by SAO specialists to work on the cryogenic cooling of optical receivers for the BTA and other applications; as a result, an extensive series (several dozen) of successfully operating cooled photodetectors was born, making us a monopolist in this field for many years. Superconductor receivers created at IRE and IAP turned out to be in demand for foreign instruments RT-22 at KrAO, which became Ukrainian (academician Valery Shulga), and the Metsahovi Observatory, Finland (Prof. Seppo Urpo). It is clear that over the years, no sub-THz antennas have appeared in Russia, and the RT25*2 radio telescope has ceased to exist. A new stage in the development of the topic was opened in the mid-2000s by A.N. Vystavkin, who proposed installing not a superheterodyne but a bolometric sub-THz receiver on the BTA. The topic of superconducting sub-THz bolometers was actively advanced within the framework of this project, but alas, it was again not completed by installing the device at the focus of the telescope. 

Finally, a new generation of researchers from the IAP and IRE have actively taken on this task. Prototype receivers have already been developed and tested in the laboratory. The problem of the ultra-deep (up to 0.3 K) cooling of such a receiver at the telescope focus is being solved, and this is also non-trivial. A Dewar with helium is not an easy thing, and the refrigerator has a compressor that generates vibrations at an unacceptable level for a telescope. However, both ideologies of sub-kelvin cooling presented above have been developed by a team of authors to a device that can be fully implemented on a telescope. 

The authors are grateful to their colleagues who have invested their labor in this long-term work and believe that it will be successfully completed. We dedicate this article to our departed colleagues: D. Korolkov, G. Chuntonov, A. Berlin, A. Vystavkin, A. Kislyakov, Yu. Lebsky, and Yu. Dryagin, and academicians Yuri Pariysky and Valery Shulga, who were involved in the origins of the development of the idea of using a sub-THz radio receiver in an optical telescope.

## Figures and Tables

**Figure 1 sensors-24-00359-f001:**
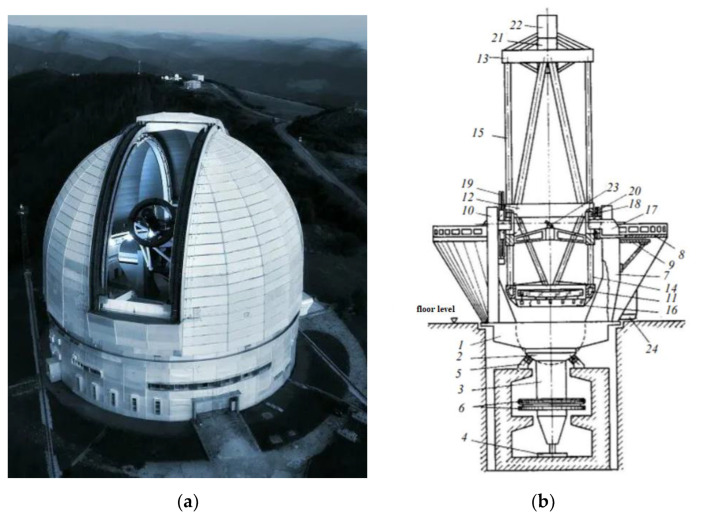
BTA: (**a**)—photo of telescope; (**b**)—design scheme: 1—platform, 2—spherical ring, 3—axis, 4—radial bearing, 5—liquid friction support, 6—wheel block, 7—rack, 8—working balconies, 9—balcony bracket, 10—elevator, 11—lower ring, 12—“mullion”, 13—upper ring, 14—lower rod, 15—upper rod, 16—main mirror frame with unloading device, 17—cantilever beams, 18—hydrostatic cushions, 19—worm gear, 20—cable drum, 21—primary focus cup, 22—observer’s cabin, 23—fixed mirror focus, 24—main stellar spectrograph.

**Figure 2 sensors-24-00359-f002:**
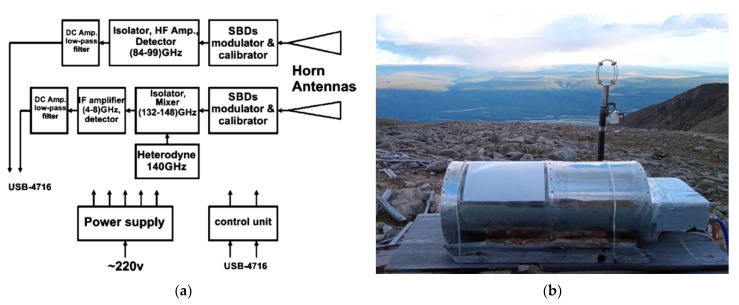
Radiometer MIAP-2: (**a**) functional scheme of the receiver unit and (**b**) a photo of the device.

**Figure 3 sensors-24-00359-f003:**
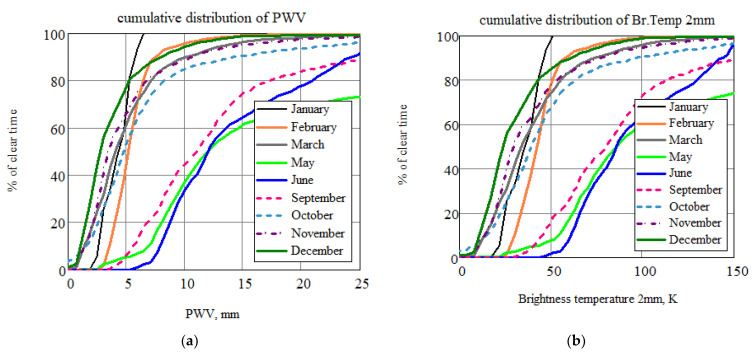
Cumulative distributions of brightness temperature (for 1.3 and 0.8 mm, the data were recalculated through water) in various atmospheric transparency windows: (**a**) 3 mm, (**b**) 2 mm, (**c**) 1.3 mm, and (**d**) 0.8 mm.

**Figure 4 sensors-24-00359-f004:**
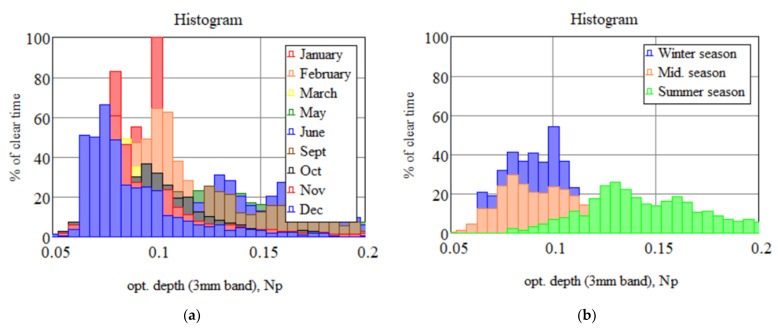
Histograms of the distribution of observation time at the BTA site by months (**a**) and seasons (**b**). Np—the neper.

**Figure 5 sensors-24-00359-f005:**
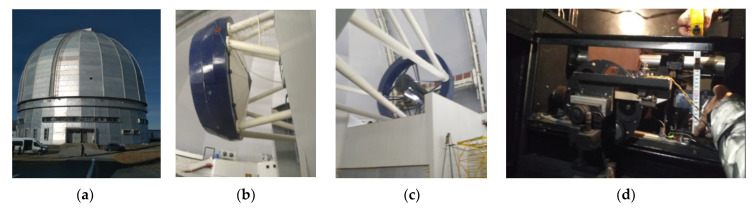
Photos from the expedition to the SAO RAS: (**a**) photo of the observatory, (**b**) photo of the six-meter mirror of the BTA telescope, (**c**) photo of the primary focus of the BTA, (**d**) measurement of landing sites in the second Nasmith focus of the BTA telescope.

**Figure 6 sensors-24-00359-f006:**
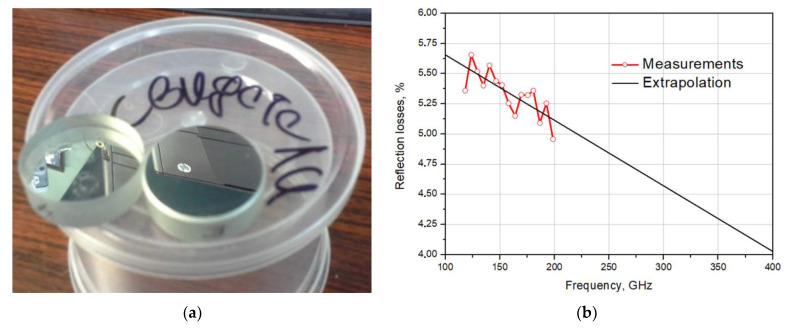
The “witness” sample: A photo of the samples (**a**) and measured reflection losses (**b**): the red graph shows measurements; the black line is extrapolation.

**Figure 7 sensors-24-00359-f007:**
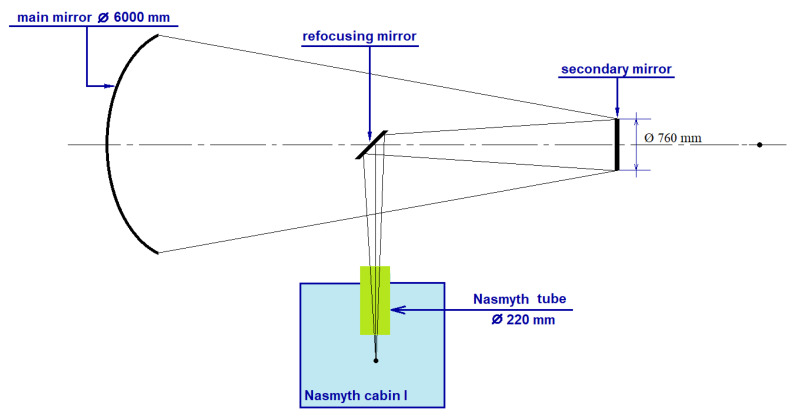
Schematic representation of the Nasmith I scheme of the BTA telescope of SAO RAS.

**Figure 8 sensors-24-00359-f008:**
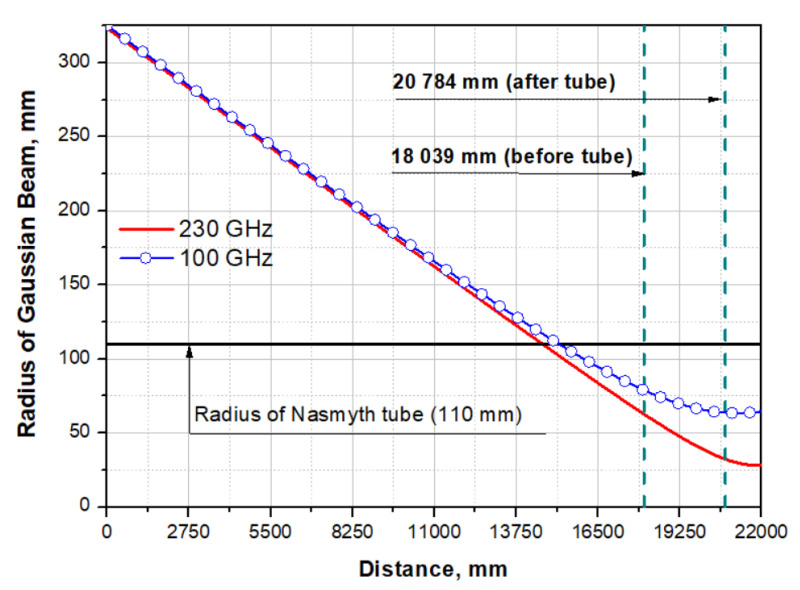
The radius of a Gaussian beam propagating in the Nasmith I scheme of the BTA telescope of the SAO RAS.

**Figure 9 sensors-24-00359-f009:**
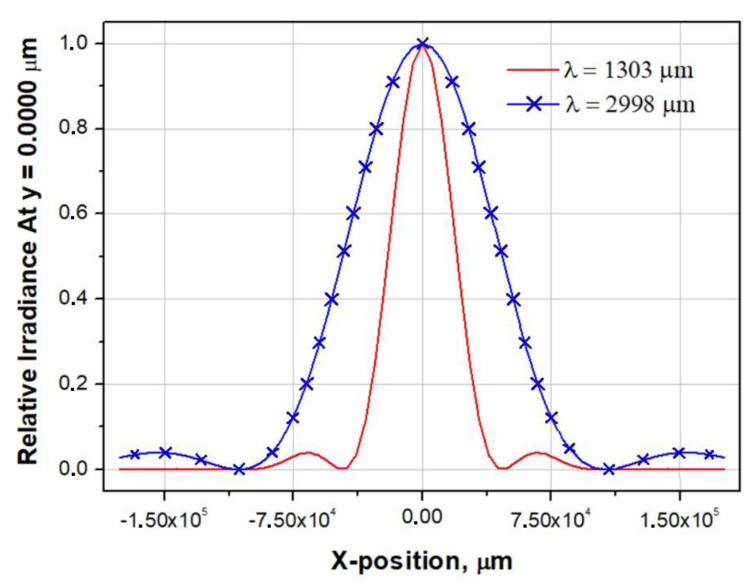
Huygens PSF cross section X for λ = 1303 μm (230 GHz) and λ = 2998 μm (100 GHz).

**Figure 10 sensors-24-00359-f010:**
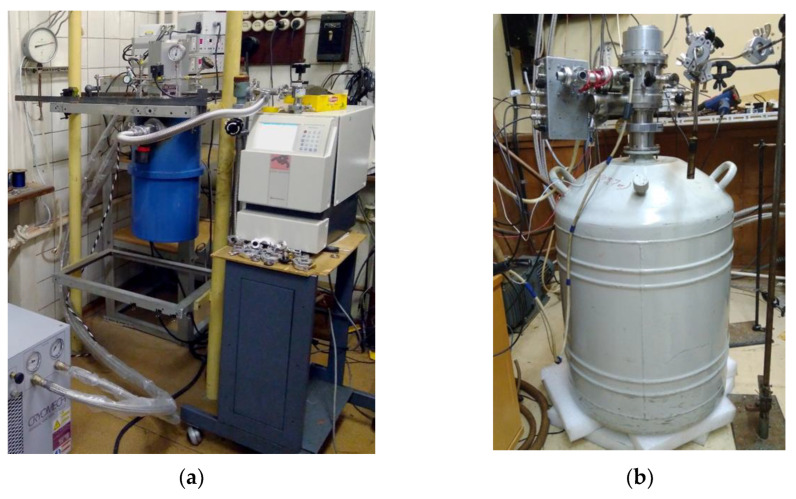
Various cryostat subK systems in which studies of the developed detection devices were carried out: (**a**) a cryostat with 3He vapor pumping Heliox AC-V closed-cycle precooling refrigerator manufactured by Oxford Instruments; (**b**) an LHe precooling cryostat with insert 3He vapor pumping stage developed at the P.L. Kapitsa Institute of Physical Problems of the Russian Academy of Sciences.

**Figure 11 sensors-24-00359-f011:**
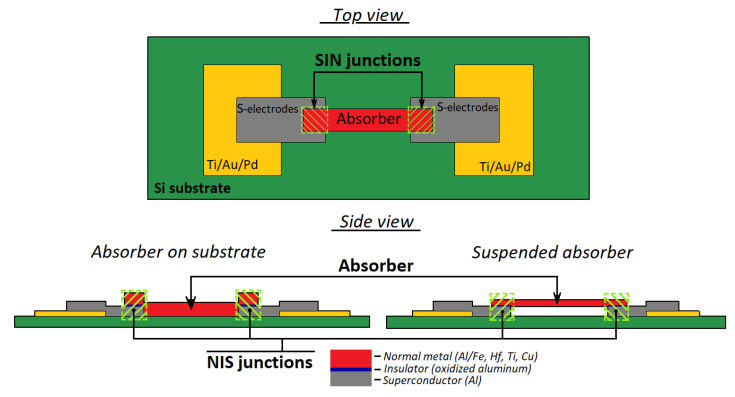
Schematic image of the SINIS structure: top view and side view of different designs of the SINIS detector, with an absorber on a substrate (left) and with a suspended absorber (right).

**Figure 13 sensors-24-00359-f013:**
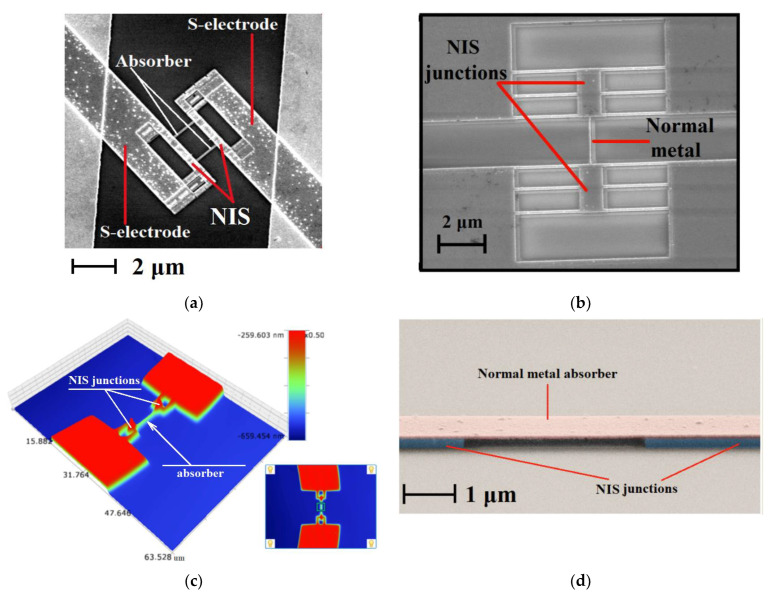
Photos of fabricated SINIS detectors: (**a**) a photograph from a scanning electron microscope of a SINIS structure made by shadow evaporation; (**b**) SINIS detector made by bridge-free technology; (**c**) SINIS detector made by separate lithography with magnetron sputtering; (**d**) a SINIS detector with a suspended absorber. Photos (**a**,**b**,**d**) are taken from [43].

**Figure 14 sensors-24-00359-f014:**
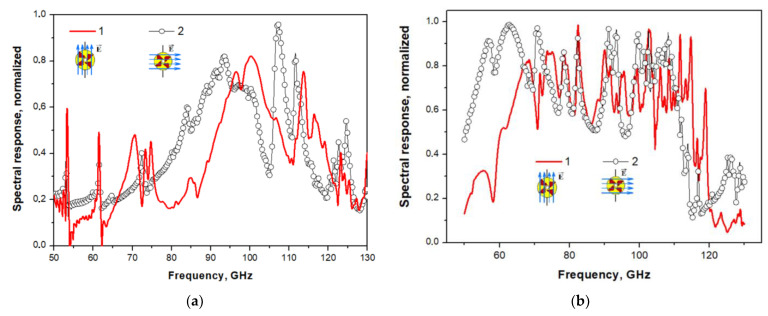
Simulation results of the spectral response of an array of annular antennas with integrated 90 GHz SINIS detectors: irradiation from the substrate (**a**) and irradiation from the antennas (**b**). Figure from [48].

**Figure 15 sensors-24-00359-f015:**
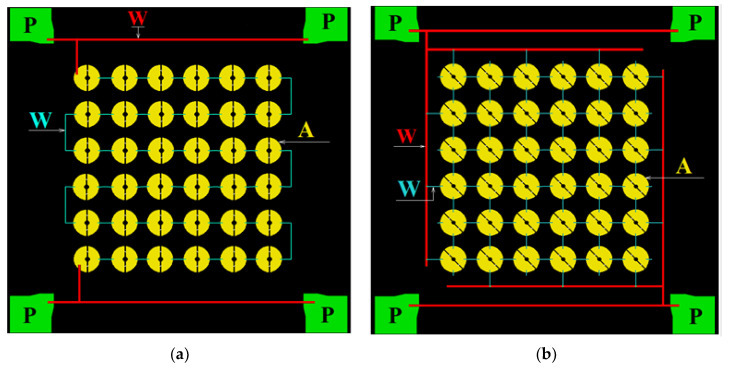
Topology of arrays with integrated SINIS detectors of the 90 GHz range: series connection of elements (**a**) and parallel connection (**b**). P—contact pads for picking up and sending a signal from/to the array; W—connecting wires (red—wide (50 μm) from the array, connecting to the contact pads; blue—narrow (2 μm) for connecting elements to each other); A—antennas, each of which have two SINIS detectors integrated.

**Figure 16 sensors-24-00359-f016:**
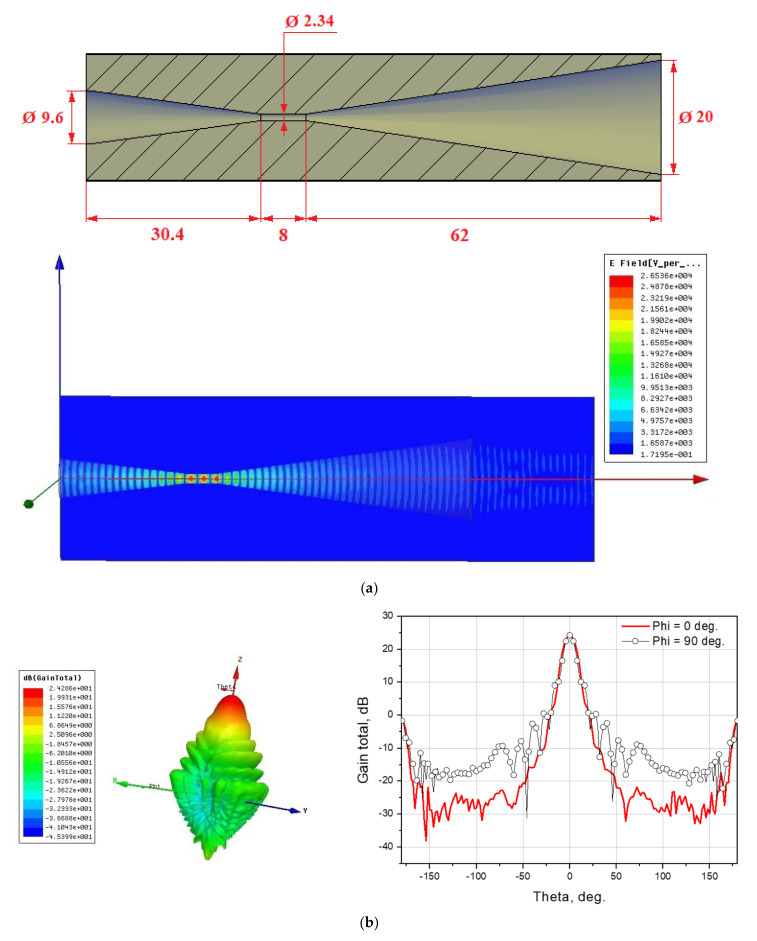
Calculated back-to-back horn for matching the receiving array with the incoming signal; dimensions are given in millimeters: (**a**) schematic representation of the horn with dimensions and (**b**) directional pattern of the horn.

**Table 1 sensors-24-00359-t001:** The radius of the Gaussian beam before and after the Nasmith tube and the fraction of the passing power.

	100 GHz	230 GHz
Before tube (18,039 mm)	79.036 mm	63.08 mm
After tube (20,784 mm)	63.638 mm	32.148 mm
At Nasmyth focus according to Zemax OpticStudio	80.29 mm	56.04 mm
Percentage of transmitted power	97.9%	99.8%

## Data Availability

Data are contained within the article.

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
