# Peer review of "Microwave Receiving System Based on Cryogenic Sensors for the Optical Big Telescope Alt-Azimuth"

_sensors, 2024, doi:10.3390/s24020359_

Round 1

Reviewer 1 Report

Comments and Suggestions for Authors

This paper reports the results of evaluating of the possibility of conducting millimeter radio astronomy with the big optical telescope at SAO in Russia. The astroclimate conditions and superconducting radio receivers are studied, the project if it is implemented will be very useful for optical and millimeter radio observations by using the same telescope, which is important for some astronomical issues. Besides, the result will be technically useful for various receivers to be mounted at the backend of optical telescopes. The paper is well organized and the results are technically introduced in details, I would like to recommend the paper can be accepted to publish. I have some further comments below.

Line 23, astroclimate or to give a full words

Line 38, you may give a full name in bracket for FIAN, KrAO

Line 63, the meaning of ‘100^m’ is not clear, please make it clearer

Line 69, astronomy in MM and subMM makes …

Line 76, you may give a full name in bracket for IPA RAS

Line 78, ‘… is positioned slightly worse’ is not clear, please rewrite it

Line 79, it’s better to remove ‘any’?

Line 82-83, ‘and over the past year … for a long time’ is not clear, you may rewrite it

In Figure 3, the caption may be …(a) 3mm. (b) 2mm, (c) 1.3mm and (d) 0.8mm ?? please check the Figure 3 (a) it is for PWV or for 3mm?

In Figure 4, you may describe what is Nep in the caption?

Line 272, to check if ‘F#=30’ is correct

Line 276, to check if ‘0,1^+’ is correct

Line 331, to check if ‘Hz^0.5’ is correct

Author Response

see attached pls

Reviewer 2 Report

Comments and Suggestions for Authors

The article suggests a project leveraging the BTA's capabilities at the SAO RAS for radio astronomical observations in the millimeter (MM) and submillimeter (subMM) range. This initiative has the potential to broaden the functionalities of the BTA optical telescope. This manuscript is well-detailed with comprehensive data and clear arguments. I believe it meets the standards for publication.

      Comments on the Quality of English Language

The article suggests a project leveraging the BTA's capabilities at the SAO RAS for radio astronomical observations in the millimeter (MM) and submillimeter (subMM) range. This initiative has the potential to broaden the functionalities of the BTA optical telescope. This manuscript is well-detailed with comprehensive data and clear arguments. I believe it meets the standards for publication.

Author Response

Thank you very much!

Reviewer 3 Report

Comments and Suggestions for Authors

This paper is important to consider the EHT project. SAO RAS shall be a great telescope to get better UV coverage with ALMA.

Author Response

Thank you very much!

Nowever improved, see corrections  green marked

Reviewer 4 Report

Comments and Suggestions for Authors

This work presents a microwave receiving system based on cryogenic sensors. In the manuscript, the breif design and experiment process and results have been expressed clearly. It is an impressive work since doing system design and implementation is really difficult and challenged. However, the manuscript still contains several concerns.

1. The title claimed that the receiving system is mainly for optical Big Telescope Alt-Azimuth. However, the main work scenario is not given. Although the authors has give two pictures in Figure 1 and 2, they are just pictures, instead of work scheme and topology. This part should be revised. 

2. Figure 3/8/12/14/16, most curves are shown in different colors but the same solid line type, this is not proper, it is much better for the authors to draw all curves in different symbols, so that to be separated much more easily.

3. Format of the entire manuscript should be improved. 

Author Response

Thank you!

Improved

See our answers and corrected manuscript with green corrections

Reviewer 5 Report

Comments and Suggestions for Authors

Dear Authors,

The manuscript is an excellent summary of your work.  I have the following suggestions:

1. Some acronyms are used before or are not defined, e.g. SAO RAS in the abstract.

2. The manuscript has some mixed formatting, e.g. on page 3.

3. Labels on graphs are sometimes hard to read, e.g. Figure 9.

I do recommend additional modifications and checks to make the manuscript even better.

Best regards.

Comments on the Quality of English Language

As per my comments to the authors.

Author Response

Thank you!

The modified version and our answers pls., find attached
